# A global seamless 1 km resolution daily land surface temperature dataset (2003-2020)

Tao Zhang[1], Yuyu Zhou[1], Zhengyuan Zhu[2], Xiaoma Li[3], Ghassem R. Asrar[4]

[1] Department of Geological and Atmospheric Sciences, Iowa State University, Ames, IA, 50011, USA

[2] Department of Statistics, Iowa State University, Ames, IA, 50011, USA

[3] School of Landscape Architecture and Art Design, Hunan Agricultural University, Changsha, Hunan, 410128, China

[4] Universities Space Research Association, Columbia, MD, 21046, USA

*Correspondence to:* Yuyu Zhou (yuyuzhou@iastate.edu)

**Abstract.** Land surface temperature (LST) is one of the most important and widely used parameters for studying land surface processes. Moderate Resolution Imaging Spectroradiometer (MODIS) LST products (e.g., MOD11A1 and MYD11A1) can provide this information with moderate spatiotemporal resolution with global coverage. However, the applications of these data are hampered because of missing values caused by factors such as cloud contamination, indicating the necessity to produce a seamless global MODIS-like LST dataset, which is still not available. In this study, we used a spatiotemporal gap-filling framework to generate a seamless global 1 km daily (mid-daytime and mid-nighttime) MODIS-like LST dataset from 2003 to 2020 based on standard MODIS LST products. The method includes two steps, 1) data pre-processing and 2) spatiotemporal fitting. In the data pre-processing, we filtered pixels with low data quality and filled gaps using the observed LST at another three time points of the same day. In the spatiotemporal fitting, first, we fitted the temporal trend (overall mean) of observations based on the day of year (independent variable) in each pixel using the smoothing spline function. Then we spatiotemporally interpolated residuals between observations and overall mean values for each day. Finally, we estimated missing values of LST by adding the overall mean and interpolated residuals. The results show that the missing values in the original MODIS LST were effectively and efficiently filled with reduced computational cost, and there is no obvious block effect caused by large areas of missing values, especially near the boundary of tiles, which might exist in other seamless LST datasets. The cross-validation with different missing rates at the global scale indicates that the gap-filled LST data have high accuracies with the average root mean squared error (RMSE) of 1.88°C and 1.33°C, respectively for mid-daytime (1:30pm) and mid-nighttime (1:30am). The seamless global daily (mid-daytime and mid-nighttime) LST dataset at a 1 km spatial resolution is of great use in global studies of urban systems, climate research and modeling, and terrestrial ecosystems studies. The data are available at Iowa State University's DataShare at https://doi.org/10.25380/iastate.c.5078492 (Zhang et al., 2021a).

## 1 Introduction

Land surface temperature (LST) is an important variable for studies of energy balance, evapotranspiration, ecosystem processes in monitoring of Earth's resources (Anderson et al., 2010; Long et al., 2020). It has been widely used in various studies such as urban heat island (Li et al., 2021b; Liu et al., 2020b; Tang et al., 2017; Yue et al., 2019), hydrology (Bai et al., 2019; Zhang et al., 2017), meteorology (Anderson et al., 2010; Li et al., 2018b), ecology (Sims et al., 2008), and energy systems (Peng et al., 2012; Zhou et

al., 2014b). LST varies significantly in both space and time due to the spatiotemporal variation of factors such as solar radiation, atmospheric conditions, land surface characteristics (Li et al., 2018a; Peng et al., 2014; Zhang et al., 2015). LST can be measured in situ, obtained from land surface modeling, and retrieved by remote sensing (Ford and Quiring, 2019; Sheffield et al., 2018). Remotely sensed LST is by far the most widely obtained/used due to its global spatial coverage, high spatiotemporal resolutions, and available long-term data records.

LST products with a variety of spatial and temporal resolutions have been developed from different sensors/satellites such as: (1) high spatial resolution of 60-120m and low temporal resolution of about every 2-16 days from Landsat (Parastatidis et al., 2017; Roy et al., 2014) and Advanced Spaceborne Thermal Emission and Reflection Radiometer (ASTER) (Hulley et al., 2015); (2) coarse spatial resolution of 3-5km but high temporal resolution sub-daily to sub-hourly from geostationary satellites (Choi and Suh, 2013; Duguay-Tetzlaff et al., 2015; Jiang and Liu, 2014; Trigo et al., 2008; Yu et al., 2009); and (3) moderate spatial resolution about 1 km and moderate temporal resolution of daily from the Moderate Resolution Imaging Spectroradiometer (MODIS) (Wan, 2013, 2014), Visible Infrared Imaging Radiometer Suite (VIIRS) (Guillevic et al., 2014), and Sea and Land Surface Temperature Radiometer (SLSTR) LST (Ghent et al., 2017). Among them, MODIS LST is the most widely used especially for regional and global studies due to its global coverage and long-term and well calibrated and documented data records (since 2000) (Aguilar-Lome et al., 2019; Li et al., 2017; Peng et al., 2012; Sandeep et al., 2021; Zhao et al., 2020b; Zhou et al., 2019). However, MODIS LST has a large number of missing values due to a variety of factors such as cloud contamination, non-overlapping satellite orbits, and instrumental malfunction (Crosson et al., 2012; Li et al., 2018a; Liu et al., 2020a; Shen et al., 2015; Wan, 2013).

Filling the missing values of MODIS LST is an effective way to overcome this limitation in MODIS LST product. Several seamless datasets have been developed in previous studies (Cheng et al., 2021; Li et al., 2018a; Metz et al., 2017; Zhang et al., 2021c; Zhao et al., 2020a), however, they only cover specific regions or have coarse spatiotemporal resolutions (Table S2). Recently, Zhan et al. (2021) produced a global 1 km LST dataset (2003 – 2019), but only a daily average of LST was included. Shiff et al. (2021) developed a Google Earth Engine (GEE) code and a web app for generating 1 km gap-filled LST by using Climate Forecast System Version 2 (CFSv2) modeled air temperature and MODIS LST data, but they did not consider the naturally spatial variation of LST. A global daily minimum and maximum LST dataset with reasonable spatial pattern that can be used for a variety of studies and applications by scientists and practitioners such as city planners and water resources managers is still not available. Meanwhile, a variety of gap-filling methods have been proposed to fill gaps in MODIS LST. These methods can be divided into four groups (Li et al., 2018a; Weiss et al., 2014; Zhang et al., 2020). The first group is based on data fusion methods, which combine LST data from different satellites or different overpasses times (e.g., morning and afternoon) of the same satellite on the same day (Crosson et al., 2012; Duan et al., 2017; Long et al., 2020; Xu and Cheng, 2021; Zhang et al., 2020, 2021b). The second group is based on empirical relationships among different methods that were used to estimate the missing values by fitting empirical relationship between LST and auxiliary data (e.g., latitude, longitude, altitude, surface moisture, normalized difference vegetation index, and ground observed LST) (Fan et al., 2014; Ke et al., 2013; Li et al., 2021a; Zhao et al., 2020a). The third group is based on the internal spatiotemporal relationship that predicted the missing values with the available LST using algorithms such as temporal interpolation (Kilibarda et al., 2014; Xu and Shen, 2013), spatial interpolation (Ke et al., 2013; Yang, 2004), spatiotemporal interpolation (Sun et al., 2017; Weiss et al., 2014), and multi-dimensional smoothing (Garcia, 2010, 2011; Liu et al., 2020a; Pham et al., 2019). The fourth group is a hybrid method that combined several methods from previous groups mentioned above (Hong et al., 2021; Li et al., 2018a; Metz et al., 2017; Weiss et al., 2014; Xu and Cheng, 2021).

However, most of the current methods have some shortcomings in accuracy and efficiency for producing globally consistent and seamless MODIS-like LST. For example, the data fusion method has the problem of mismatch between LST from different sources and usually cannot fully fill gaps (Crosson et al., 2012). The computational cost of the methods based on the empirical

relationship could increase significantly with the increase of spatial resolutions and might not be able to fully capture spatial and temporal variations of LST as the auxiliary data have low temporal resolutions (Fan et al., 2014; Ke et al., 2013; Zhao et al., 2020a). The temporal interpolation and multi-dimensional smoothing methods are computationally efficient but may miss short-term temporal variations of LST (Kilibarda et al., 2014; Xu and Shen, 2013). The spatial interpolation methods may lead to physically unrealistic features in the interpolated LST when there are a lot of missing observations (Ke et al., 2013; Yang, 2004). The spatiotemporal interpolation methods can capture the short-term changes of LST but are time-consuming due to the use of local moving windows for each pixel (Li et al., 2018a; Weiss et al., 2014). The hybrid methods take the advantages of the methods mentioned above and carry with it some of shortcomings of these methods, and may actually amplify them in the process of merging data imputed using different methods.

We proposed a spatiotemporal gap-filling framework to gap-fill missing values in MODIS LST product with good accuracies and high efficiencies. This framework includes two key steps of preprocessing and spatiotemporal fitting. Based on this framework, we developed a global 1 km daily (mid-daytime and mid-nighttime) LST dataset from 2003 to 2020 using the 1 km daily MODIS LST product. The remainder of this paper describes the study area and data (Sect. 2), the proposed spatiotemporal gap-filling approach (Sect. 3), the results and discussion (Sect. 4), data availability (Sect. 5), and conclusions (Sect. 6).

## 2 Study area and data

The study area is nearly the entire global land surface, including 178 MODIS tiles (Fig. 1). The 1 km daily MODIS LST product Version 6 from 2003 to 2020 is the primary data used in this study. It was produced based on the National Aeronautics and Space Administration (NASA) Earth Observing System (EOS) satellites Terra and Aqua (MOD11A1 and MYD11A1) (Wan, 2013, 2014). There are four observations each day from the two satellites (i.e., 10:30 am and 10:30 pm for Terra (T1 and T3), 1:30 am and 1:30 pm for Aqua (T2 and T4)). Another two auxiliary datasets used are the annual MODIS land cover product (MCD12Q1) (Sulla-Menashe and Friedl, 2018) and urban extents derived from nighttime light observations and their surrounding rural areas (Zhou et al., 2014a, 2018). Water pixels from the MCD12Q1 product were excluded in our analysis.

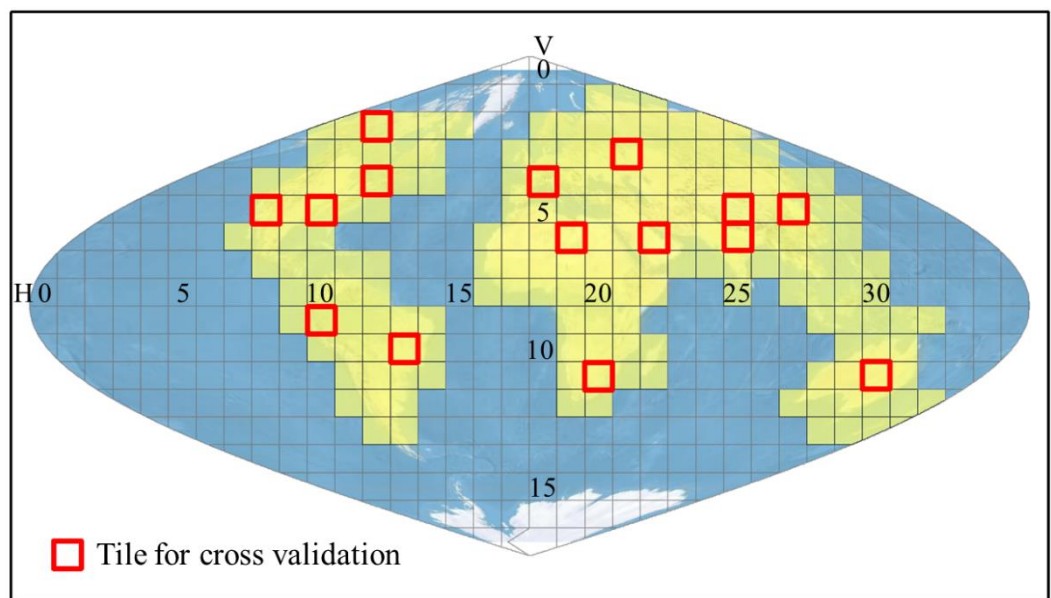

**Figure 1: MODIS data tiles used in gap-filling and cross validation analysis.**

## 3 Method

We developed a spatiotemporal gap-filling framework to gap-fill missing values in MODIS daily LST to produce a seamless 1 km spatial resolution global dataset from 2003 to 2020 (Fig. 2). The framework includes two key steps, 1) data pre-processing (Sect. 3.1) and 2) spatiotemporal fitting (Sect. 3.2). This gap-filling method was applied to MODIS LST at T2 (~1:30pm, Aqua Day in Fig. 2) and T4 (~1:30am, Aqua Night in Fig. 2), respectively, to build the 1 km daily LST (maximum (mid-daytime) and minimum (mid-nighttime)) data. In the sections below, we described each of these steps in detail.

**3.1 Data pre-processing**

Data pre-processing includes two parts: 1) data filtering and 2) daily merge. We first checked the quality of original MODIS data based on its quality assurance (QA) information and removed data points with error > 3K. We applied this threshold value because a stricter (or lower) value can exclude most of LST in urban areas (Crosson et al., 2012; Metz et al., 2017). Second, we conducted a daily merge using four observations from the two satellites (Terra and Aqua) in a given day using a modified algorithm from Li

et al. (2018a). Taking a pixel with missing value of T2 as an example (Fig. 2), we calculated percent of valid data (PVD) in a year for all four observations, respectively. When PVD of T2 is smaller than 5% and one PVD of T1, T4, or T3 is greater than 5%, we gap-filled missing values of T2 using data from one of the other three observations based on the order of T1, T4, and T3. If PVD of T1 is greater than 5%, we estimated T2 by T1 using the linear regression method with T2 as dependent variable and T1 as independent variable based on available time series of LSTs in a year. If PVD of T4 is greater than 5%, we estimated T2 by T4

using the shift method (i.e., adding T4 and adjusted daily difference between T2 and T4 to get T2). If PVD of T3 is greater than 5%, we estimated T2 by T3 using the shift method (i.e., adding T3 and adjusted daily difference between T2 and T3 to get T2). After the daily merge, we gap-filled the left missing values using the spatiotemporal fitting. We selected the threshold of PVD as 5% because the valid data smaller than 5% is not enough to capture the spatial pattern of LST in a tile according to our experiments. Details of the linear regression and shift methods can be found in Li et al. (2018a). Specifically, we used the shift method because

there is non-linear relationship between daytime and nighttime LSTs (i.e., T2 and T3 (or T4)) (Crosson et al., 2012). We estimated the daily shift using temporally interpolating monthly averaged shift, i.e., monthly mean LST difference between T2 and T3 (or T4), and then we added the daily shift to T3 (or T4) to estimate T2.

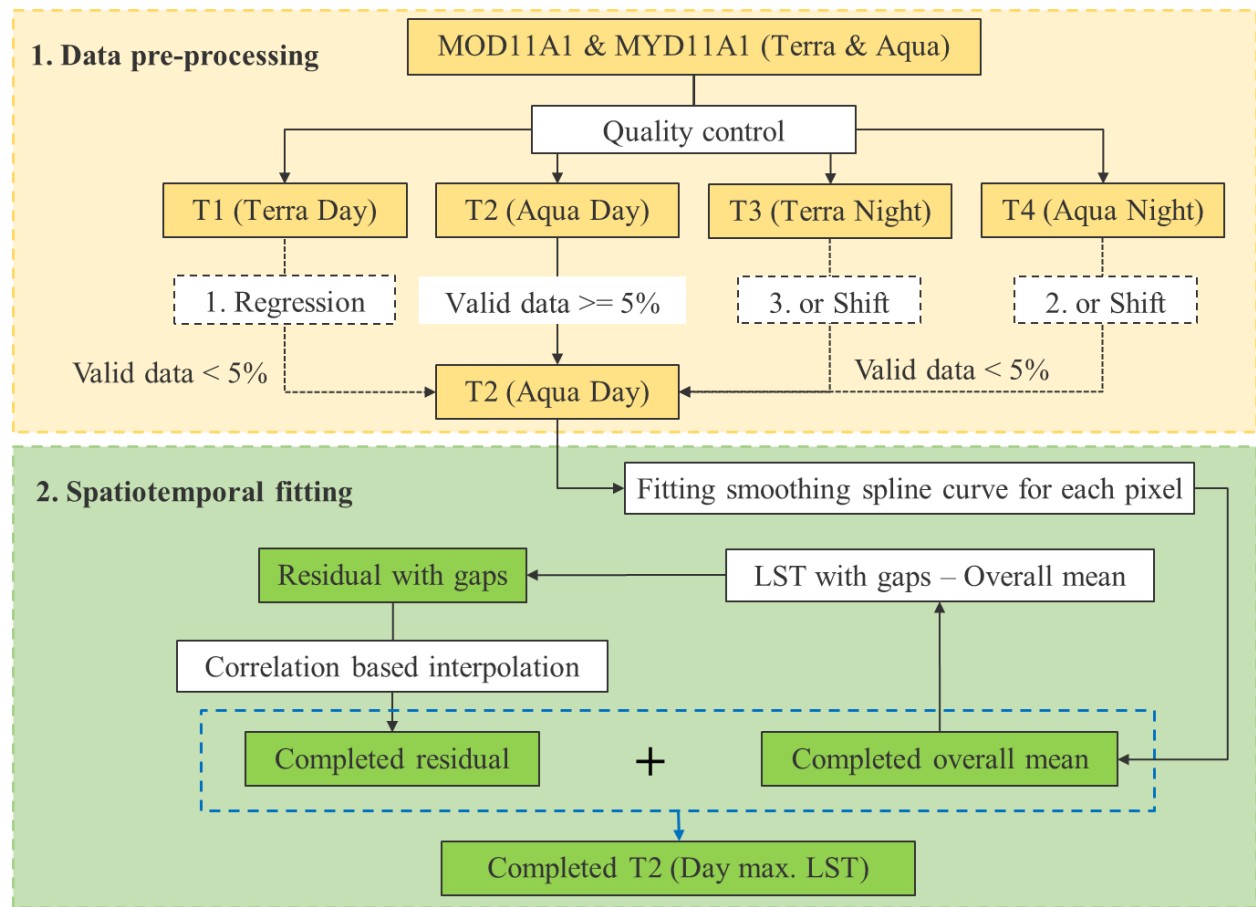

**Figure 2: An overview of the spatiotemporal gap-filling framework (Taking T2 as an example).**

### 3.2 Spatiotemporal fitting

The spatiotemporal fitting algorithm includes three steps (Fig. 3). First, we fitted the overall mean of observations in each pixel (i.e., the fitted daily values (temporal trend) in a year using the smoothing spline function for which the independent variable is the day of year) using a smoothing spline function (Green and Silverman, 1994) to capture the overall trend. Specifically, the overall means of T2 and T4 were independently estimated. The time series of daily LST in a year (e.g., LST of T2) can be divided into two components, the overall mean (trend) and daily residual with gaps (daily fluctuations). We used the smoothing spline function for fitting overall trend since this algorithm does not have hypothesis on the shape of the seasonal trend and is capable to capture different seasonal patterns of LST across the globe. Second, we spatiotemporally interpolated residuals for each day using a correlation-based method (Details in Sect. 3.2.1), in which the missing residual of a target pixel was estimated based on the temporally and linearly regressive correlation between target pixel and its 8 neighboring valid pixels (i.e., with good quality). We used the daily residuals of a year from target pixel and its neighboring pixels to estimate the missing values. When the value of the target pixels is missing for a specific day, we can still build linear regression functions based on the time series data. We selected 1% of the uniformly distributed pixels (10 km intervals) as representative neighboring pixels to perform the interpolation of residuals with high efficiency without reducing the accuracy based on our experiments. Moreover, we divided the global land surface area into 9 overlapped zones to avoid possible boundary effects (Details in Sect. 3.2.2). Finally, the seamless overall mean and daily residuals were added to obtain the gap-filled LST data.

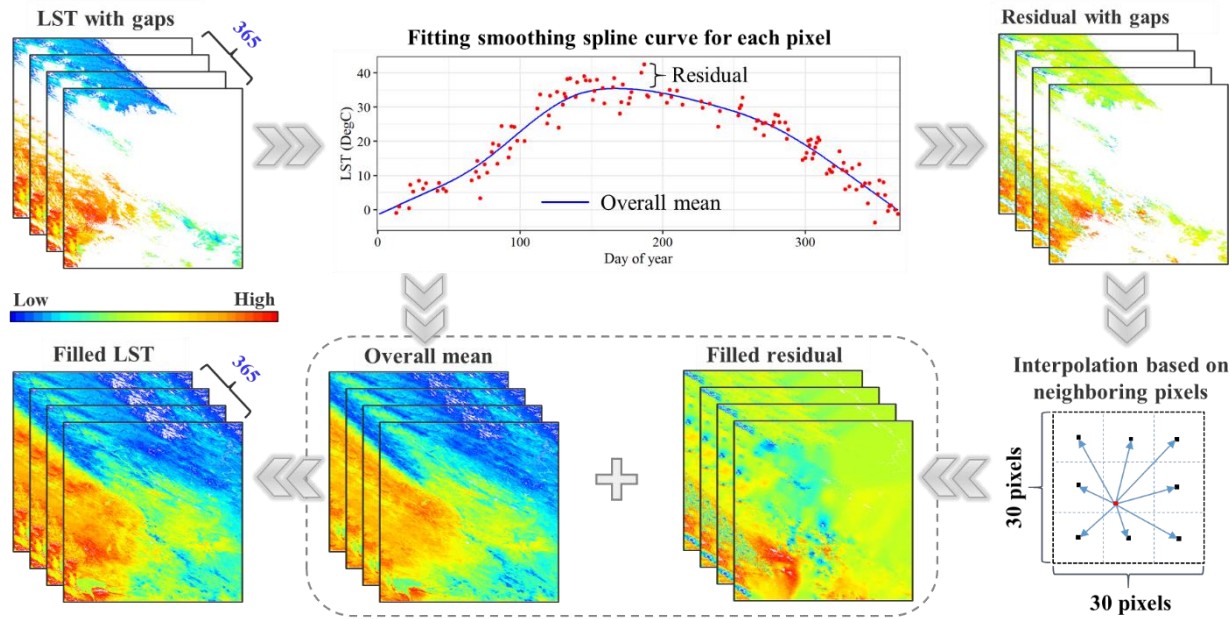

**Figure 3: An example of the spatiotemporal fitting algorithm for gap-filling LST.**

### 3.2.1 Interpolation based on Correlation Weighting (ICW)

An Interpolation based on Correlation Weighting (ICW) technique was used to interpolate the residual of land surface temperature (LST) in each day of a year. This method was inspired by the Inverse Distance Weighting (IDW) interpolation method. The IDW method uses the weighted average values of neighboring sites to estimate the missing value, where the weight was calculated based on the inverse distances between target site and its neighboring sites. In the ICW method, the weight between target site and one of its neighboring sites was calculated based on the correlation between the time series of the pairs (days of a year) LSTs of the two locations.

The missing value of the target site $V_{S_0}$ at the time $t$ was estimated based on values of the neighboring sites with Eq. (1).

$$V_{S_0}(t) = \sum_{i=1}^{i=n} w(S_0, S_i) \cdot V_{S_0}(S_i, t) \tag{1}$$

where $V_{S_0}(t)$ is the estimated value of target site at the time $t$; $w(S_0, S_i)$ is the weight of the $i$-th neighboring site $S_i$, which can be calculated with Eq. (2); $V_{S_0}(S_i, t)$ is the estimated value of the target site at the time $t$ based on the $i$-th neighboring site, which can be estimated with Eq. (3); and $n$ is the number of neighboring sites.

$$w(S_0, S_i) = \frac{cor(S_0, S_i)}{\sum_{i=1}^{i=n} cor(S_0, S_i)} \tag{2}$$

where $w(S_0, S_i)$ and $n$ are the same with those in Eq. (1); $cor(S_0, S_i)$ is the Pearson's correlation coefficient between $S_0$ and $S_i$.

$$V_{S_0}(S_i, t) = \alpha_i + \beta_i \cdot V_{S_i}(t) \tag{3}$$

where $\alpha_i$ and $\beta_i$ are the intercept and slope of the linear function between target site and the $i$-th neighboring site, which was fitted using ordinary least square (OLS) method based on the matched time series (days of a year) of LST in the two locations; $V_{S_i}(t)$ is the value of the $i$-th neighboring sites at time $t$.

### 3.2.2 Implementation of the ICW method

The ICW method was implemented as follows. First, in order to improve the efficiency, each MODIS tile was divided into blocks with a size of 10 by 10 pixels, and the block center pixels were used as neighboring pixels for interpolating missing residuals. That is, missing residuals in a block can be interpolated based on the values from the 8 neighboring block center pixels. Second, in order to ensure that all the block center pixels have valid (good quality) data for the estimation of other pixels, the missing values in the block center pixels were interpolated using the IDW method. The steps used in this process are: (a) computing the average value of each block; (b) resampling the original MODIS tile of 1200 by 1200 to 120 by 120 and the value of each pixel in the new image is the average value of a block in the original MODIS tile; (c) interpolating missing values in the resampled image based on the IDW method; and (d) assigning interpolated values to the block center pixels without valid values in the original MODIS tile. Third, in order to reduce the possible boundary effects of the interpolated residuals between neighboring blocks, for each pixel of a block, one of the neighboring block center pixels, which has the largest correlation coefficient with the target pixel was used for estimation. This process can avoid systematic deviation in the boundary pixels from different blocks that were estimated based on different combination of block center pixels, because all the pixels in a block were interpolated based on eight center pixels of neighboring blocks. Eqs. (1) and (3) can be simplified to Eq. (4).

$$V_{S_0}(t) = \alpha_m + \beta_m \cdot V_m(t) \tag{4}$$

where $\alpha_m$ and $\beta_m$ are the intercept and slope of the linear function between target pixel and its neighboring pixel with the corresponding maximum correlation coefficient, which was fitted using the ordinary least square (OLS) method based on the matched time series (days of a year) values of the two locations; and $V_m(t)$ is the value of the neighboring pixel with the maximum correlation coefficient at the time $t$.

Finally, in order to mitigate boundary effects between neighboring tiles, multiple neighboring tiles were mosaicked as a region, and residual of the block center pixels in the region were interpolated at the same time. The overlapped areas between the two neighboring regions were also considered to avoid possible boundary effects. The interpolation was conducted following the order of the region ids (Fig. 4). For example, as the id of North America is 1, which was the first region to be processed.

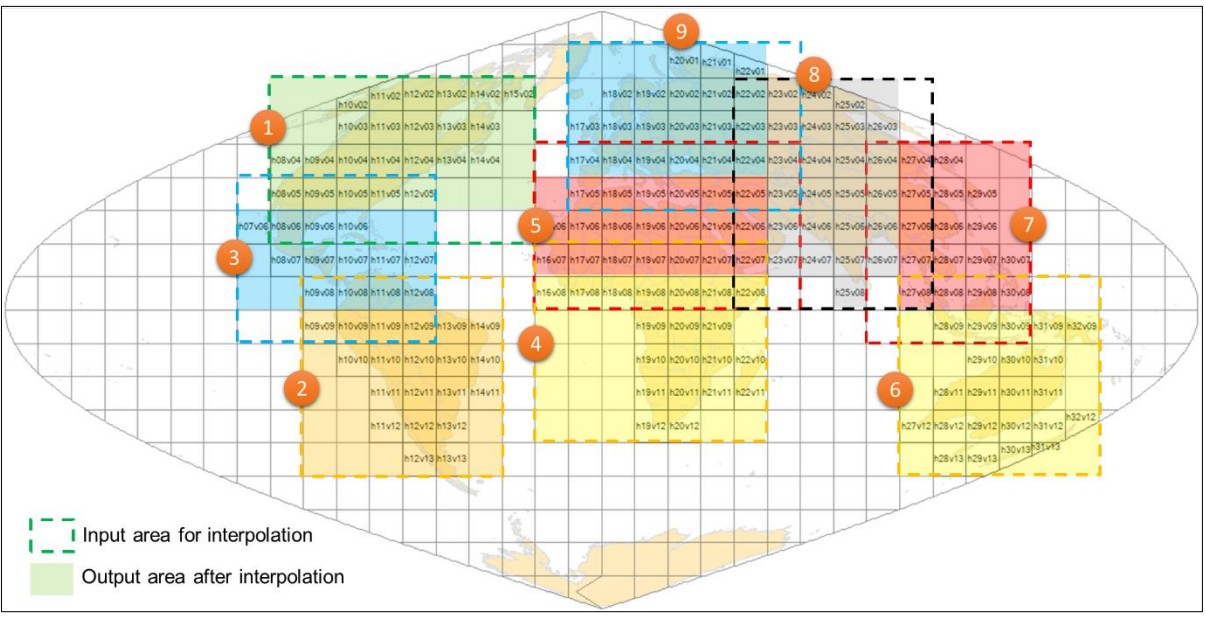

**Figure 4: Division of global regions. Dashed and shaded rectangles indicate the extent of input data and output data, respectively.**

### 3.3 Accuracy assessment

We evaluated the accuracy of the gap-filled data using cross validation by randomly selecting 15 MODIS tiles in 2005, 2010, and 2015 (Fig. 1). In each year, we selected 19 days with the maximum observations of high quality data (i.e., daily data with valid observations larger than 95% percentile in a year) in the cross validation. For each of the selected days, we manually introduced gaps under three scenarios (i.e., excluding 25%, 50%, and 75% of valid pixels) based on the spatial pattern of missing pixels from another day of the year. Then we gap-filled these missing values and compared them with the original values. We calculated root mean square error (RMSE) as the indicator of accuracy (Eq. (5)).

$$RMSE = \sqrt{\frac{1}{n}\sum_{i=1}^{n}\left(\widehat{LST_i} - LST_i\right)^2} \tag{5}$$

where $LST_i$ and $\widehat{LST_i}$ are original MODIS LST and gap-filled LST values of the $i$-th pixel; $n$ is the number of the gap-filled pixels.

### 4 Results and discussion

#### 4.1 Accuracy of gap-filled LST

The results of cross validation indicate the gap-filled LST has high accuracies (Fig. 5 and Table 1). The observed and gap-filled LSTs of representative pixels for different ratios of exclusion scattered along the 1:1 line with RMSE ranging from 2.05 to 2.31 °C and from 1.35 to 1.62 °C, respectively for daytime and nighttime (Fig. 5). As shown in Table 1, the average RMSE at tile level (i.e., calculating based on all the excluded pixels of each tile) ranges from 1.20 to 2.13 °C with an average of 1.88°C and 1.33°C, respectively for daytime and nighttime. The lowest RMSE occurs in 2010 with 25% excluding rate for nighttime, while the highest RMSE occurs in 2015 with 75% excluding rate for daytime. Compared with the accuracies at tile level, the accuracies for urban areas (i.e., calculating based on urban pixels in the excluded areas of each tile) are always higher with RMSE ranging from 1.14 to 2.06 °C (Table 1).

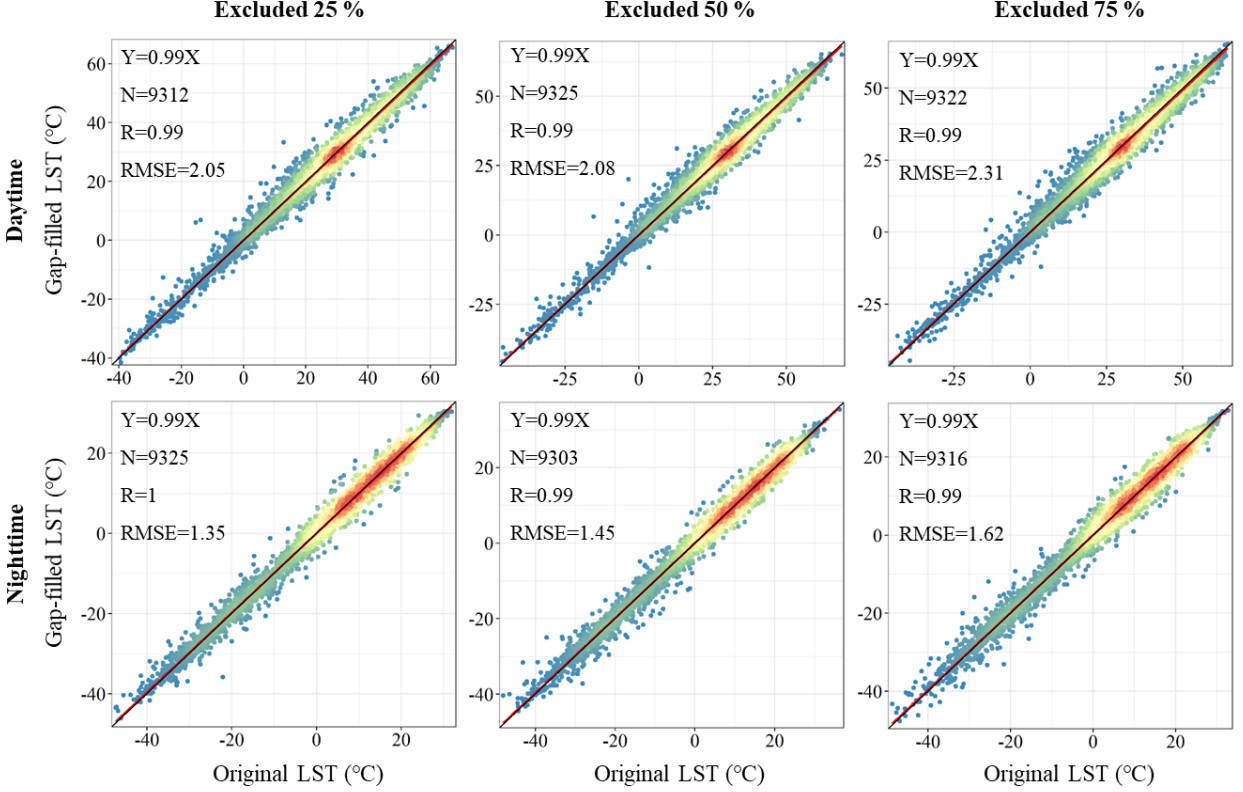

**Figure 5: Scatter plots between gap-filled LST and original MODIS LSTs for daytime and nighttime in the excluded areas used for cross validation. We used 855 images (15 tiles × 3 years × 19 days) and selected 11 pixels from the excluded area in each image in the scatter plots. Meanwhile, we excluded values of water pixels in accuracy assessment. The color of points represents the density of points, where the red points represent the highest density, and the blue points represent the lowest density. The red solid line represents the regression line, and the black line is the 1:1 line.**

**Table 1. Average RMSEs of 15 tiles used in cross-validation analysis of efficacy of the gap-filling method (Unit: ℃)**

| Time | Year | RMSE in excluded area (±standard deviation) | | | | | |
|---|---|---|---|---|---|---|---|
| | | 25% | | 50% | | 75% | |
| | | Tile level | Urban area | Tile level | Urban area | Tile level | Urban area |
| Daytime | 2005 | 1.77 (±0.41) | 1.68 (±0.46) | 1.78 (±0.44) | 1.74 (±0.55) | 2.06 (±0.38) | 2.01 (±0.52) |
| | 2010 | 1.74 (±0.49) | 1.67 (±0.71) | 1.76 (±0.50) | 1.69 (±0.74) | 1.91 (±0.52) | 1.87 (±0.69) |
| | 2015 | 1.90 (±0.53) | 1.82 (±0.72) | 1.91 (±0.54) | 1.95 (±0.64) | 2.13 (±0.55) | 2.06 (±0.61) |
| Nighttime | 2005 | 1.21 (±0.36) | 1.15 (±0.35) | 1.30 (±0.35) | 1.23 (±0.35) | 1.45 (±0.35) | 1.42 (±0.44) |
| | 2010 | 1.20 (±0.30) | 1.14 (±0.32) | 1.29 (±0.29) | 1.29 (±0.39) | 1.43 (±0.35) | 1.38 (±0.44) |
| | 2015 | 1.28 (±0.42) | 1.19 (±0.33) | 1.37 (±0.39) | 1.25 (±0.36) | 1.48 (±0.41) | 1.47 (±0.44) |

Note: 'Urban' means the urban and their surrounding rural areas. The tile level means the accuracy was calculated based on all the excluded pixels of each tile; urban area means the accuracy was calculated based on urban pixels in the excluded areas of each tile. Each RMSE value is the mean of RMSEs from all selected days in 15 selected MODIS tiles.

When the number of missing values in original LST increases, the gap-filled LST data tends to reduce in accuracy (Fig. 5, Table 1). As shown in Fig. 5, when the excluding rate increases from 25% to 75%, the RMSE of LST for daytime and nighttime increases from 2.05 to 2.31℃ and 1.35 to 1.62℃ for daytime and nighttime, respectively. This is also true across all years at the tile level and in urban area in Table 1. However, the RMSE values are still within reasonable ranges. When the excluding rate is 75%, the RMSEs are 2.31℃ and 1.62℃, respectively for daytime and nighttime (Fig. 5). Meanwhile, 88.9% of the RMSE in Table 1 is lower than 2 ℃. Besides, the accuracies of gap-filled LST vary with climate zones and may be also correlated with landforms (Table S1).

## 4.2 Spatial and temporal patterns of LST

The examples of global LST data illustrate that the missing values in the original MODIS LST have been effectively gap-filled using the proposed gap-filling algorithm (Fig. 6). In the original MODIS LST, the continuously missing values mainly occur in Eastern Asia, South Asia, and Central Africa for both daytime and nighttime, in the example date (Fig. 6). In the gap-filled data, the missing values in these regions were fully gap-filled.

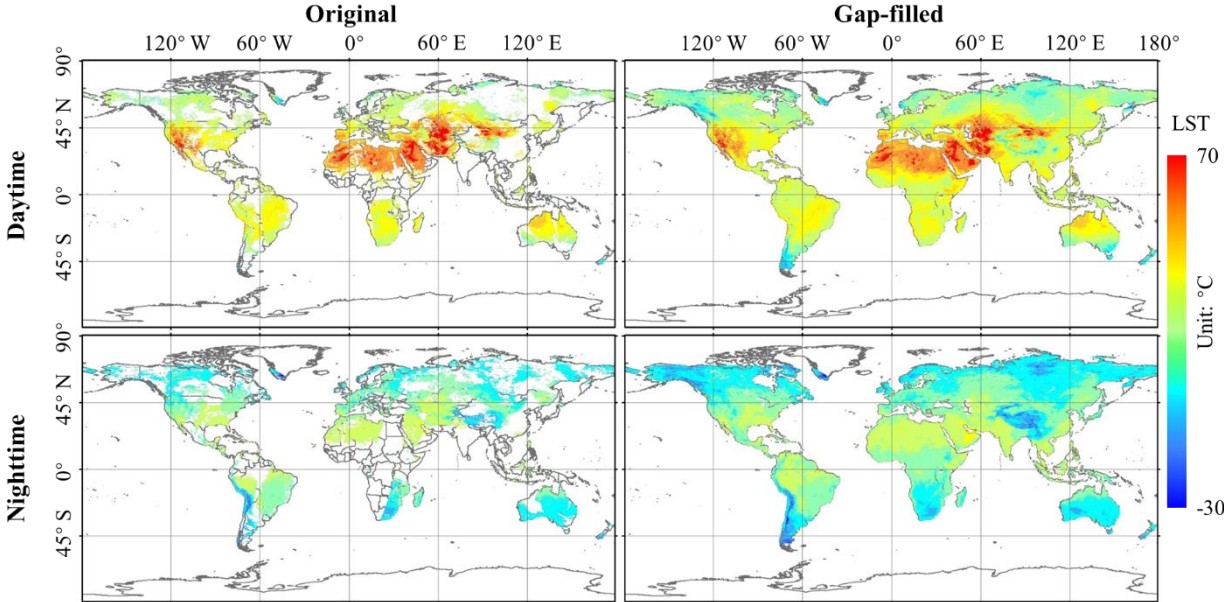

**Figure 6:** Spatial pattern of original and gap-filled LSTs at global scale on day 200 of year 2020.

The comparisons of spatial patterns between gap-filled and original MODIS LST in representative cities around the world (Fig. 7) illustrate that the missing values in the original MODIS LST have been effectively gap-filled at the city scale. As shown in Fig. 7, there is no missing value in the entire land surface area of the gap-filled data (water pixels were masked as NA). The gap-filled data capture well urban heat island (UHI) phenomenon (i.e., high temperature in urban than that of the surrounding rural areas). The spatial pattern of the gap-filled LST is reasonable with transition from urban to rural areas and there are no obvious boundary

effects (more details in Sect. S1 and Sect. S2). For example, there is no obvious boundary effect between two MODIS tiles in the gap-filled LST data in Huston area, which suggests the interpolation of residuals (Sect. 3.2.2) in the proposed method is reliable. The gap-filled LST in the Pearl River Delta region shows a number of small speckles because this region is an agglomeration of sub-areas undergoing rapid urbanization.

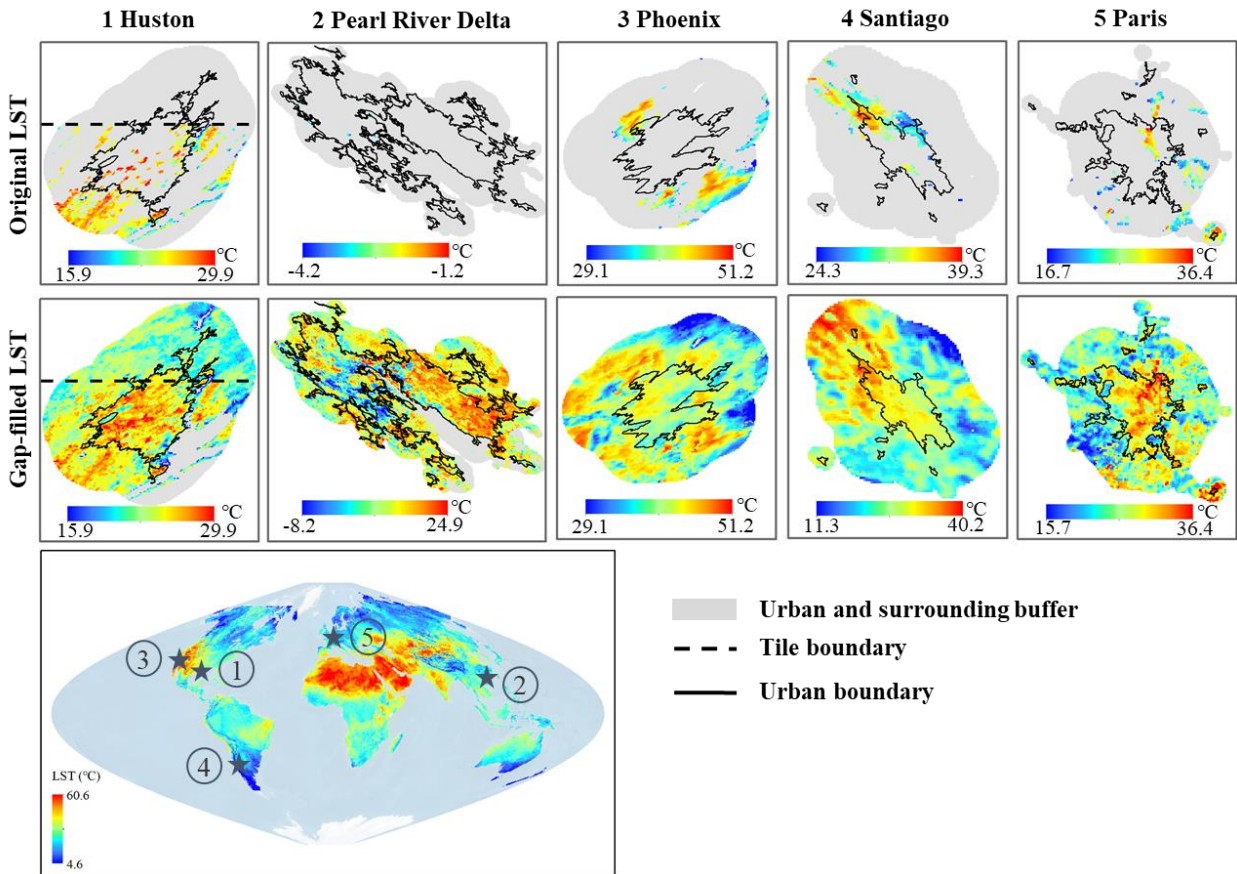

**Figure 7: Spatial pattern of original and gap-filled LSTs in five representative cities. NA (gray color) in the gap-filled LST is water pixels. Black color solid lines are the boundary of urban regions extracted by using global artificial impervious area data with 30 m spatial resolution (Li et al., 2020).**

The comparison of temporal pattern between gap-filled and original MODIS LSTs in a mega-city (Fig. 8) illustrates that the missing data in the original MODIS LST can be effectively and completely gap-filled for the entire period. As shown in Fig. 8, there are several days with limited valid (high quality) observations in original MODIS LST in Beijing, China in daytime in 2010, and these missing values were fully gap-filled in our data for the entire period. When there are only a few missing values in original LST data (days 28 and 130 in Fig. 8), the gap-filled and original LSTs show similar spatial pattern with significant UHI phenomenon. When there are large number of missing values in original LST data (days 219 and 293 in Fig. 8), the gap-filled LSTs can also illustrate the UHI phenomenon but different LST magnitudes with that of the former cases. Therefore, we may get more accurate estimation of annual average LST based on the gap-filled LST than original LST data.

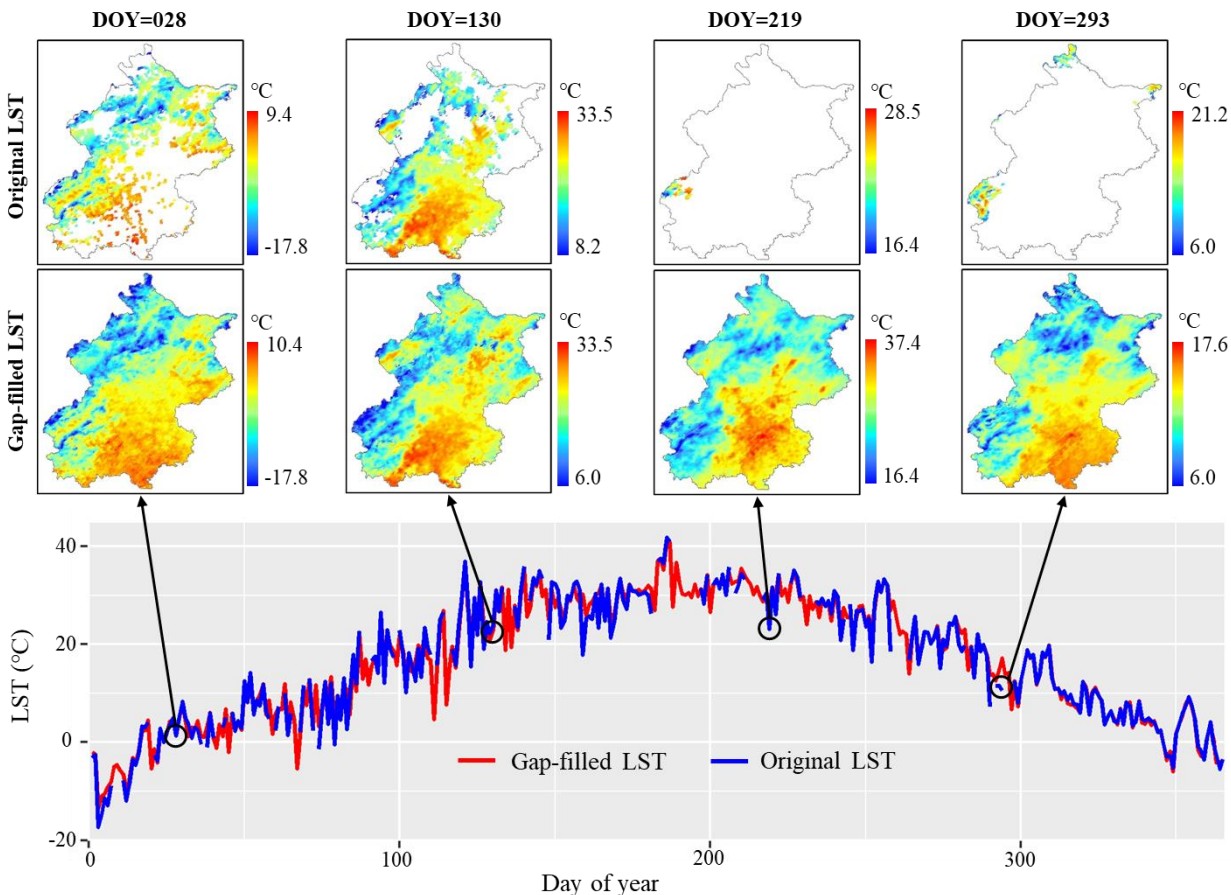

**Figure 8: Temporal pattern of average daytime LST from original and gap-filled data in Beijing in year 2010. The black circles are example days showing maps of original and gap-filled LST data.**

### 4.3 Comparison with existing seamless LST data

255 The accuracy of the resulting gap-filled LST from this study is comparable or better when compared with other reported seamless LST datasets. Our gap-filled LST data shows higher accuracies compared with the gap-filled LSTs based on the hybrid spatiotemporal gap-filling method proposed by Li et al. (2018a). These two datasets are most comparable because of the use of similar accuracy evaluation method (cross validation at the global scale) in both studies. In the hybrid method proposed by Li et al. (2018a), about 11% to 60% of the valid values (personal communication) were excluded for cross validation purpose in the

260 urban areas at the global scale, and the average RMSE is 3.29℃ and 2.68℃, for daytime and nighttime, respectively. In this study, the average RMSE is 1.83℃ and 1.28℃ in the urban and surrounding areas for daytime and nighttime, respectively (Table 1). The gap-filled LSTs based on the data fusion method implemented on GEE (Shiff et al., 2021) were also evaluated at the global scale, but the mean RMSE is 2.7℃, higher than that of this study. The accuracies of other seamless LST datasets were generally evaluated based on a limited number of in-situ LST observations (Zhang et al., 2019; Zhou et al., 2017), which are not exactly the same as

265 satellite LSTs (Hong et al., 2021), and the evaluation in these studies are not directly comparable with our study. For example, the LST data by Zhao et al. (2020) reached the average RMSE of 1.59℃ at the daily level; the LST data by Zhang et al. (2021c) showed the RMSE ranging from 2.03K to 3.98K in Tibetan Plateau (Zhang et al., 2021b); the LST data using a hybrid method (Hong et al., 2021) has a mean absolute error (MAE) of 1.0K at the daily level (Table S2).

 The gap-filled LST in this study does not have the issue of boundary effect that might exist in the previous methods. Li et al.

270 (2018a) combined several techniques including data fusion (Crosson et al., 2012), spatiotemporal interpolation (Gerber et al., 2018; Weiss et al., 2014), and temporal interpolation methods (Xu and Shen, 2013) to reconstruct daily (mid-daytime and mid-nighttime)

LST. The systematic differences between neighboring regions with the use of different gap-filling techniques in the hybrid method may lead to boundary effects (Li et al., 2018a). The data fusion method implemented on GEE (Shiff et al., 2021) directly filled the missing values in MODIS LST using the estimated LST values without consideration of the spatial continuity, which might lead to boundary effects. The seamless LST data produced by Zhao et al. (2020) might also contain boundary effects since different regression methods were used to reconstruct the missing values according to the number of valid pixels. There are no obvious boundary effects in the LST data by Zhang et al. (2021c) using the data fusion model proposed by Zhang et al. (2021b). However, abrupt changes might occur between the original valid MODIS LST and the gap-filled LST using the data fusion model (Figs. 7 and 8 in the study by Zhang et al. (2021b)). The gap-filled LST data in this study using the novel framework consisting of two key steps (Sect. 3.2.2) can mitigate boundary effects between neighboring regions (Fig. S1), neighboring tiles (Fig. S2), and within a given tile (Fig. 7 and Sect. S1).

The gap-filled global 1km daytime and nighttime LST data have advantages regarding spatiotemporal resolutions (i.e., daily minimum and maximum) or coverage (i.e., global) and have significant potential for use in many disciplines of Earth system science and applications (Table S2). In the existing seamless LST datasets, Zhan et al. (2021) produced a global daily average 1 km resolution LST dataset from 2003 to 2019, without resolving by daytime and nighttime. Zhao et al. (2020a) developed monthly average LST with 5.6km spatial resolution for China from 2003 to 2017. Cheng et al. (2021) published a daily (mid-daytime and mid-nighttime) 1 km seamless LST of China from 2002 to 2020. Zhang et al. (2021c) generated a daily (daytime and nighttime) 1 km all-weather LST dataset for China and its surrounding areas for 2000 to 2020. Li et al. (2018a) produced a 1 km daily (mid-daytime and mid-nighttime) LST dataset only in urban and surrounding rural areas of United States. Shiff et al. (2021) only provided GEE code for producing global 1 km daily (mean, mid-daytime and mid-nighttime) LST data. The LST data in this study have a spatial resolution of 1 km and include daily LST at mid-daytime and mid-nighttime with a global coverage from 2003 to 2020, which has higher spatiotemporal resolutions or coverage than other existing published seamless LST datasets.

The gap-filling framework proposed in this study can be efficiently implemented and has advantages regarding computing time compared to other algorithms/methods. For example, the gap-filling method proposed by Zhao et al. (2020a) were used for monthly 5.6km resolution LST data, and it may require significant computation time for higher spatiotemporal resolution (daily, 1 km) LST data because it needs to calculate the distance between similar valid pixels and each target pixel (with missing or low quality value) based on a geographically weighted regression method. The gap-filling method proposed by Zhang et al. (2021b) is also complex and time consuming due to involvement of multi-source data and complex parameterization process on a pixel-by-pixel basis. The daily average LST data produced by Zhan et al. (2021) were calculated based on the nonlinear annual temperature cycle (ATC) and diurnal temperature cycle (DTC) modelling on a pixel-by-pixel basis, which is time-consuming for global scale applications (Hong et al., 2021). The hybrid gap-filling method proposed by Li et al. (2018a) is time consuming due to the use of spatiotemporal interpolation (Gerber et al., 2018; Weiss et al., 2014) algorithm, in which the missing value of a pixel at a specific time and location was interpolated by using a quantile regression in the corresponding local spatial and temporal window. In the proposed method in this study, the interpolation of the residual for a pixel at a specific time was implemented by calculating correlation coefficients and fitting linear regression functions using the time series data of the target pixel and its neighboring pixels in the corresponding local window (Sect. 3.2.1). Moreover, 1% of pixels at central pixels of blocks ($10 \times 10$ pixels) were used as neighboring pixels for interpolation of residual (Sect. 3.2.2) to reduce the amount of calculation, and the relevant parameters (i.e., correlation coefficients and coefficients of linear regression function) between target pixel and its 8 nearest neighboring pixels were calculated only one time for the entire period (365 days for a year) based on the time series of residuals. The reason is that the time series of residuals from two neighboring pixels within a short distance are highly correlated with each other. Our scheme can significantly improve the efficiency for global applications without reducing the accuracy according to our experiments.

The accuracy of the gap-filled LST should not be significantly affected by the land cover type and elevation differences in local spatial windows. LST values from different land cover types and elevations within a small spatial region may be significantly different (Zhang et al., 2021b). These differences of LST values can be captured though the temporal pattern of LST (overall mean)
by separately fitting the smoothing spline curves (Fig. 3), and the spatiotemporal similarity of residuals between neighboring pixels were gap-filled. The gap-filled LST values are the sum of overall mean and residuals. Therefore, our method can capture the missing values of LST in different land cover types and elevations in local spatial windows.

A limitation of this study is that the gap-filled LST dataset mainly reflects the clear-sky conditions and future work can focus on recovering cloudy-sky LST to produce all-weather LST dataset when high-quality ancillary data become available. As only the
spatiotemporal information of clear-sky MODIS LST data was used to fill the missing values, the gap-filled pixels mainly reflect the clear-sky LST and might overestimate the actual LST values. Previous studies have attempted to develop methods for obtaining all-weather LST data by incorporating cloudy-sky LST retrieved from passive microwave observations or reanalyzed products (Duan et al., 2017; Long et al., 2020; Zhang et al., 2019, 2020, 2021b) or adding clear-sky LST and the LST differences resulting from cloud impacts according to the surface energy balance (SEB) methods (e.g., Jia et al., 2021). However, it is challenging to
obtain cloudy-sky LST and cloud caused LST differences at the global scale in the last two decades because the ancillary datasets have lower spatial resolutions and accuracies compared to MODIS LST, leading to complicated algorithms with complicated hypotheses (Long et al., 2020; Zhang et al., 2021b). Future studies are needed to develop robust and efficient algorithms for producing global all-weather LST data.

## 5 Data availability

Data described in this manuscript can be accessed at Iowa State University's DataShare at https://doi.org/10.25380/iastate.c.5078492 (Zhang et al., 2021a). The dataset contains 36 sub datasets (one for each year in daytime and nighttime from 2003 to 2020). Each sub dataset contains LST data of a specific time (daytime or nighttime) and specific year (2003 – 2020) and is organized by day of year. The data are in GeoTIFF with the georeferenced information embedded. Each file keeps the MODIS Ellipse Sinusoidal projection with a spatial resolution of 1 km. The unit of LST in Geotiff is 0.1
Celsius temperature (0.1 $^{o}$C), and the naming rule can be found in the file of "README.pdf".

## 6 Conclusions

We propose a framework for filling the gap in long-term Earth observations and geophysical data records that are used by many Earth system science disciplines and applications. We used the proposed method to generate a globally consistent and 1 km daily (mid-daytime and mid-nighttime) MODIS-like LST data from 2003 to 2020 using MODIS LST datasets (MOD11A1 and
MYD11A1), which has advantages in spatial coverage and spatiotemporal resolutions compared to existing studies. The resulting dataset filled all existing gaps resulting from elimination of poor-quality data seamlessly with high accuracies based on a cross validation under different rates of missing values for both daytime and nighttime. The average RMSE of gap-filled LST for daytime and nighttime ranges from 1.80 to 2.03 $^{o}$C and 1.23 to 1.45 $^{o}$C, respectively, when different percentages of the data were excluded. The results show that the missing values in the original MODIS LST were effectively and efficiently filled, and there is no obvious
block effect caused by large areas of missing values, especially near the boundary of tiles, which might exist in other seamless LST datasets. The gap-filled global 1 km daily LST dataset can provide better data source for multidisciplinary applications such as urban heat island, air temperature estimation, soil moisture estimation, evapotranspiration, and drought monitoring (Phan and Kappas, 2018). However, it is worth noting that the accuracy of the gap-filled LST can be influenced by the rate of missing values,

indicating that uncertainties might increase with the increase of missing values in the original dataset. Moreover, future work can

focus on diurnal changes of LST by increasing observations within a day.

**Supplement.**

**Author contributions.**

Yuyu Zhou designed the research; Tao Zhang implemented the research and wrote the original manuscript; Yuyu Zhou and

Zhengyuan Zhu supervised the research. All co-authors revised the manuscript and contributed to the writing.

**Competing interests.**

The authors declare that they have no conflict of interest.

**Acknowledgements.**

This research was supported by the College of Liberal Arts and Science's (LAS) Dean's Emerging Faculty Leaders
award at the Iowa State University and the National Science Foundation (2041859).

**Financial support.**

**Review statement.**

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
