# Peer review of "A global seamless 1 km resolution daily land surface temperature dataset (2003-2020)"

_Earth System Science Data, 2021_

## Author Comment (AC1)

**Reviewer #1:**

This paper proposes a very interesting spatiotemporal gap-filling framework to generate seamless global 1 km daily (mid-daytime and mid-nighttime) LST products from 2003 to 2020. The manuscript is clearly written and well-structured. The products provide global coverage and better accuracies for long-term time-series LSTs. After incorporating my suggested edits, I recommend acceptance.

**Response:** Thank you very much for your suggestions. Below please find our responses to your comments in detail.

**Major comments**

**1.** The novelty of this work is not comprehensively presented throughout the study. The global significance and necessity of this work should be clarified further. It can be more clearly presented in the abstract, introduction, and conclusion sections.

**Response:** Thank you very much for your suggestion. We emphasized the novelty in the abstract (line 13-15), introduction (line 52-60), and conclusion (line 342) sections in the revised manuscript.

"*However, the applications of these data are hampered because of missing values caused by factors such as cloud contamination, indicating the necessity to produce a seamless global MODIS-like LST dataset, which is still not available.*" (line 13-15).

"*Several seamless datasets have been developed in previous studies (Cheng et al., 2021; Li et al., 2018a; Metz et al., 2017; Zhang et al., 2021c; Zhao et al., 2020a), however, they only cover specific regions or have coarse spatiotemporal resolutions (Table S2). Recently, Zhan et al. (2021) produced a global 1 km LST dataset (2003 – 2019), but only a daily average of LST was included. Shiff et al. (2021) developed a Google Earth Engine (GEE) code and a web app for generating 1 km gap-filled LST by using Climate Forecast System Version 2 (CFSv2) modeled air temperature and MODIS LST data, but they did not consider the naturally spatial variation of LST. A global daily minimum and maximum LST dataset with reasonable spatial pattern that can be used for a variety of studies and applications by scientists and practitioners such as city planners and water resources managers is still not available.*" (line 52-60).

"*which has advantages in spatial coverage and spatiotemporal resolutions compared to existing studies*" (line 340).

**2.** A more detailed explanation of the methodology is necessary. I suggest that SI and S2 should be moved to Section 3.2 (spatiotemporal fitting).

**Response:** Thanks for your suggestion. We have moved S1 and S2 to Section 3.2 and changed them as Section 3.2.1 and Section 3.2.2. Details are shown in line 143-186 in the revised manuscript. The numbers of corresponding figures were also revised in the rest of the revised manuscript and in the supplement.

**3.** More comparisons to previous methods are necessary for the discussion section to show the validity of your method, except for Li et al. (2018).

**Response:** Thanks for your suggestion. We have added relevant discussion of comparisons in the revised manuscript.

"*The gap-filled LSTs based on the data fusion method implemented on GEE (Shiff et al., 2021) were also evaluated at the global scale, but the mean RMSE is 2.7℃, higher than that of this study.*" (line 261-263).

"*For example, the LST data by Zhao et al. (2020) reached the average RMSE of 1.59℃ at the daily level; the LST data by Zhang et al. (2021c) showed the RMSE ranging from 2.03K to 3.98K in Tibetan Plateau (Zhang et al., 2021b); the LST data by using a hybrid method (Hong et al., 2021) has a mean absolute error (MAE) of 1.0K at the daily level (Table S2).*" (line 265-268).

"*The data fusion method implemented on GEE (Shiff et al., 2021) directly filled the missing values in MODIS LST using the estimated LST values without consideration of the spatial continuity, which might lead to boundary effects. The seamless LST data produced by Zhao et al. (2020) might also contain boundary effects since different regression methods were used to reconstruct the missing values according to the number of valid pixels. There are no obvious boundary effects in the LST data by Zhang et al. (2021c) using the data fusion model proposed by Zhang et al. (2021b). However, abrupt changes might occur between the original valid MODIS LST and the gap-filled LST using the data fusion model (Figs. 7 and 8 in the study by Zhang et al. (2021b)).*" (line 273-279).

**4.** The method may help reconstruct seamless global daily LST products based on MODIS datasets, but the study is limited on clear-sky conditions. How about the LSTs under cloudy

conditions? More discussions may be needed.

**Response:** Thank you again for your suggestion. We added a paragraph to discuss it in line 318-328.

"*A limitation of this study is that the gap-filled LST dataset mainly reflects the clear-sky conditions and future work can focus on recovering cloudy-sky LST to produce all-weather LST dataset when high-quality ancillary data become available. As only the spatiotemporal information of clear-sky MODIS LST data was used to fill the missing values, the gap-filled pixels mainly reflect the clear-sky LST and might overestimate the actual LST values. Previous studies have attempted to develop methods for obtaining all-weather LST data by incorporating cloudy-sky LST retrieved from passive microwave observations or reanalyzed products (Duan et al., 2017; Long et al., 2020; Zhang et al., 2019, 2020, 2021b) or adding clear-sky LST and the LST differences resulting from cloud impacts according to the surface energy balance (SEB) methods (e.g., Jia et al., 2021). However, it is challenging to obtain cloudy-sky LST and cloud caused LST differences at the global scale in the last two decades because the ancillary datasets have lower spatial resolutions and accuracies compared to MODIS LST, leading to complicated algorithms with complicated hypotheses (Long et al., 2020; Zhang et al., 2021b). Future studies are needed to develop robust and efficient algorithms for producing global all-weather LST data.*" (line 318-328).

**Minor Comments**

**1.** Line 21 and Line 237-242: Compared with other gap-filling methods, further details should be provided as to how to quantify the effectiveness and efficiency of the gap-filling framework.

**Response:** Thanks for your suggestion. We have improved the description in the revised manuscript.

"*The results show that the missing values in the original MODIS LST were effectively and efficiently filled with reduced computational cost*" (line 22-23).

"*In the proposed method in this study, the interpolation of the residual for a pixel at a specific time was implemented by calculating correlation coefficients and fitting linear regression functions using the time series data of the target pixel and its neighboring pixels in the corresponding local window (Sect. 3.2.1). Moreover, 1% of pixels at central pixels of blocks (10 × 10 pixels) were used as neighboring pixels for interpolation of residual (Sect. 3.2.2) to reduce the amount of calculation, and the relevant parameters (i.e., correlation coefficients and*

*coefficients of linear regression function) between target pixel and its 8 nearest neighboring pixels were calculated only one time for the entire period (365 days for a year) based on the time series of residuals.*" (line 303-309).

**2.** Line 94-95: confusing sentence, please rephrase.

**Response:** Thank you for pointing out. We have modified it as "*Another two auxiliary datasets used are the annual MODIS land cover product (MCD12Q1) (Sulla-Menashe and Friedl, 2018) and urban extents derived from nighttime light observations and their surrounding rural areas (Zhou et al., 2014a, 2018).*"

**3.** Line 111: What are the criteria for identifying the PVD at 5%.

**Response:** Thank you for your question. We clarified in the revised manuscript.

"*We selected the threshold of PVD as 5% because the valid data smaller than 5% is not enough to capture the spatial pattern of LST in a tile according to our experiments.*" (line 117-119).

**4.** I understand Section 3.1 is from previous studies, but it is necessary to provide more basic information about this section.

**Response:** Thanks for your suggestion. We have added more details in the revised manuscript.

"*When PVD of T2 is smaller than 5% and one PVD of T1, T4, or T3 is greater than 5%, we gap-filled missing values of T2 using data from one of the other three observations based on the order of T1, T4, and T3. If PVD of T1 is greater than 5%, we estimated T2 by T1 using the linear regression method with T2 as dependent variable and T1 as independent variable based on available time series of LSTs in a year. If PVD of T4 is greater than 5%, we estimated T2 by T4 using the shift method (i.e., adding T4 and adjusted daily difference between T2 and T4 to get T2). If PVD of T3 is greater than 5%, we estimated T2 by T3 using the shift method (i.e., adding T3 and adjusted daily difference between T2 and T3 to get T2). After the daily merge, we gap-filled the left missing values using the spatiotemporal fitting. We selected the threshold of PVD as 5% because the valid data smaller than 5% is not enough to capture the spatial pattern of LST in a tile according to our experiments. Details of the linear regression and shift methods can be found in Li et al. (2018a). Specifically, we used the shift method because there is non-linear relationship between daytime and nighttime LSTs (i.e., T2 and T3 (or T4)) (Crosson et al., 2012). We estimated the daily shift using temporally interpolating monthly averaged shift, i.e.,*

*monthly mean LST difference between T2 and T3 (or T4), and then we added the daily shift to T3 (or T4) to estimate T2.*" (line 111-122).

**5.** Fig 2 should be revised.

**Response:** Thank you for your suggestion. We improved this figure.

**6.** Put more details for Fig 3.

**Response:** Thank you for your suggestion. We have added more details for this figure.

**7.** What are the advantages of using the smoothing spline function?

**Response:** Thank you for your question. We have clarified it in the revised manuscript.

"*We used the smoothing spline function for fitting overall trend since this algorithm does not have hypothesis on the shape of the seasonal trend and is capable to capture different seasonal patterns of LST across the globe.*" (line 131-133).

**8.** I did not see any consideration about PVD<5% of four observations from the two satellites.

**Response:** Thank you for your suggestion. We have clarified it in the revised manuscript.

"*After the daily merge, we gap-filled the left missing values using the spatiotemporal fitting.*" (line 117).

**9.** Line 138-139: Not clear of how you manually introduced under three scenarios gaps here? Are the three gaps randomly distributed or continuously missing? How was the reconstruction accuracy in different distributions?

**Response:** Thank you for your questions. We manually add gaps based on the spatial pattern of missing pixels from another day of the year. Therefore, the distribution of the gaps is not randomly, and we considered the reconstruction accuracy under different missing ratios. We have improved the description in line 190-192.

"*For each of the selected days, we manually introduced gaps under three scenarios (i.e., excluding 25%, 50%, and 75% of valid pixels) based on the spatial pattern of missing pixels from another day of the year.*" (line 190-192).

**10.** Why does the number of pixels (N range from 9300 to 9350) vary approximately among

different excluded areas (25%, 50%, and 75%)?

**Response:** Thank you for your question. The number of pixels among different excluded areas are different. In order to draw scatter plots in figure 4 (figure 5 in the revised manuscript), we used 855 images (15 tiles × 3 years × 19 days) and selected 11 pixels from the excluded area in each image in the plotting. The selected number of pixels are same in these scatter plots. In addition, we excluded values of water pixels for accuracy assessment. The caption of figure 5 was improved in the revised manuscript.

"*Figure 5: Scatter plots between gap-filled LST and original MODIS LSTs for daytime and nighttime in the excluded areas used for cross validation. We used 855 images (15 tiles × 3 years × 19 days) and selected 11 pixels from the excluded area in each image in the scatter plots. Meanwhile, we excluded values of water pixels in accuracy assessment. The color of points represents the density of points, where the red points represent the highest density, and the blue points represent the lowest density. The red solid line represents the regression line, and the black line is the 1:1 line.*" (line 207-211)

**11.** Line 155: I consider the selection of 10 pixels per image seems to be not reasonable for cross-validation analysis.

**Response:** Thank you for your comment. All the excluded values were used for evaluating the accuracy, as shown in Table 1. We randomly selected 11 pixels (double checked the code) per image for a better visualization in the scatter plot. It was clarified in the caption of figure 5.

**12.** Line 214 does not provide a citation?

**Response:** Thank you for pointing out that. We have added the reference.

"*The systematic differences between neighboring regions with the use of different gap-filling techniques in the hybrid method may lead to boundary effects (Li et al., 2018a).*" (line 272-273).

**13.** Line 219-224: I saw this very similar sentence in the introduction.

**Response:** Thank you for your suggestion. We have improved it in the introduction in the revised manuscript.

"*Filling the missing values of MODIS LST is an effective way to overcome this limitation in MODIS LST product. Several seamless datasets have been developed in previous studies (Cheng et al., 2021; Li et al., 2018a; Metz et al., 2017; Zhang et al., 2021c; Zhao et al., 2020a), however,*

*they only cover specific regions or have coarse spatiotemporal resolutions (Table S2). Recently, Zhan et al. (2021) produced a global 1 km LST dataset (2003 – 2019), but only a daily average of LST was included. Shiff et al. (2021) developed a Google Earth Engine (GEE) code and a web app for generating 1 km gap-filled LST by using Climate Forecast System Version 2 (CFSv2) modeled air temperature and MODIS LST data, but they did not consider the naturally spatial variation of LST.*" (line 52-58).

**Reviewer #2:**

Zhang et al. proposed a novel method to generate a seamless global 1 km daily MODIS-like LST dataset. The topic is very interesting and the performance of the method looks good. The newly developed dataset should be of great use in climatic and ecological studies. I would like recommend acceptance after following minor revisions.

**Response:** Thank you very much for your suggestions. Below please find our responses to your comments in detail.

**Line 15:** "first, we fitted the long-term trend (overall mean) of observations in each pixel (ordered by day of year)" is confusing. First, the "overall mean" is ambiguous. Second, the independent variable should be included (e.g., Day of year?). Last, the fitting method should be introduced as well.

**Response:** Thank you for your suggestions. First, the "overall mean" comes from the statistics community. So, we kept it in the manuscript, but clarified it in detail. We have improved this sentence as "*first, we fitted the temporal trend (overall mean) of observations based on the day of year (independent variable) in each pixel using the smoothing spline function*" (line 19-20).

**Lines 40-42:** Some descriptions are not so accurate. For example, the spatial resolution of the Landsat series and ASTER data should be 60-120m; the temporal resolution for the ASTER data is not 16 days; the MODIS data should not be termed as "high spatial resolution about 1 km and high temporal resolution of…." as compared to the former two groups of the datasets. May be "moderate" is more appropriate.

**Response:** Thank you for your suggestions. We have clarified them in the revised manuscript.
"*(1) high spatial resolution of 60-120m and low temporal resolution of about every 2-16 days from Landsat (Parastatidis et al., 2017; Roy et al., 2014) and Advanced Spaceborne Thermal Emission and Reflection Radiometer (ASTER) (Hulley et al., 2015); (2) coarse spatial resolution of 3-5km but high temporal resolution sub-daily to sub-hourly from geostationary satellites (Choi and Suh, 2013; Duguay-Tetzlaff et al., 2015; Jiang and Liu, 2014; Trigo et al., 2008; Yu et al., 2009); and (3) moderate spatial resolution about 1 km and moderate temporal resolution of daily from the Moderate Resolution Imaging Spectroradiometer (MODIS) (Wan, 2013, 2014), Visible Infrared Imaging Radiometer Suite (VIIRS) (Guillevic et al., 2014), and Sea and Land Surface Temperature Radiometer (SLSTR) LST (Ghent et al., 2017).*" (line 41-47).

**Line 57:** It remains unclear what are the so-called "there are still limitations in these products" here. It seems that this paragraph list some previous studies without pointing out the specific knowledge gaps. Maybe it's better to incorporate this paragraph into the following one.

**Response:** Thank you for the suggestion. We have combined the two paragraphs and emphasized the knowledge gaps of previous data products in the revised manuscript.

"*Several seamless datasets have been developed in previous studies (Cheng et al., 2021; Li et al., 2018a; Metz et al., 2017; Zhang et al., 2021c; Zhao et al., 2020a), however, they only cover specific regions or have coarse spatiotemporal resolutions (Table S2). Recently, Zhan et al. (2021) produced a global 1 km LST dataset (2003 – 2019), but only a daily average of LST was included. Shiff et al. (2021) developed a Google Earth Engine (GEE) code and a web app for generating 1 km gap-filled LST by using Climate Forecast System Version 2 (CFSv2) modeled air temperature and MODIS LST data, but they did not consider the naturally spatial variation of LST. A global daily minimum and maximum LST dataset with reasonable spatial pattern that can be used for a variety of studies and applications by scientists and practitioners such as city planners and water resources managers is still not available.*" (line 52-60).

**Line 75:** I think the method based on empirical relationship is not necessarily computationally expensive as compared to the spatiotemporal interpolation and hybrid methods. For example, I came across studies that filled the missing LST values based on the ERA5-Land skin temperature might be more efficiently though with low accuracies.

**Response:** Thank you for the comments. We agree that when the spatial resolution is coarse (e.g., ERA5 data), it is not time-consuming to use regression methods. With the increase of spatial resolution and the number of explanatory variables, the computational cost could increase significantly. We have improved relevant descriptions in in the revised manuscript.

"*The computational cost of the methods based on the empirical relationship could increase significantly with the increase of spatial resolutions and might not be able to fully capture spatial and temporal variations of LST as the auxiliary data have low temporal resolutions (Fan et al., 2014; Ke et al., 2013; Zhao et al., 2020a).*" (line 74-76).

**Line 97:** The legends of figure 1 are somewhat confusing. I think this study filled the gaps for all the tiles. Thus the "Title for gap-filling" should be removed to avoid possible misleading.

**Response:** Thank you for your suggestion. As suggested, we removed "Tile for gap-filling".

**Line 102-108:** Most descriptions here are completely the same as that in Abstract. Given that these have been detailed in the following sections, it's better to simplify or remove the duplications.

**Response:** Thank you for the suggestion. We have deleted the description of the calculation steps of the method.

**Line 115:** "If PVD of T2 >= 5%, did not change the data and accepted it, otherwise we gap-filled missing values based on the following order": I am lost on whether this study filled the missing values or not for the "PVD of T2 >= 5%".

**Response:** Thank you for your question. We clarified it in the revised manuscript. We did not fill the missing values when "PVD of T2 is larger than 5%" in the data pre-processing step. We filled the missing values in the spatiotemporal fitting step. We have explained it as "*After the daily merge, we gap-filled the left missing values using the spatiotemporal fitting.*" (line 117).

**Lines 116-118:** Did this study fills the missing values by all the three methods or just one of them for a given pixel location?

**Response:** Thank you for your question. Only one method was used for a given pixel location. Only if the missing value was not filled, the following method will be used. We have clarified the description as "*When PVD of T2 is smaller than 5% and one PVD of T1, T4, or T3 is greater than 5%, we gap-filled missing values of T2 using data from one of the other three observations based on the order of T1, T4, and T3. If PVD of T1 is greater than 5%, we estimated T2 by T1 using the linear regression method with T2 as dependent variable and T1 as independent variable based on available time series of LSTs in a year. If PVD of T4 is greater than 5%, we estimated T2 by T4 using the shift method (i.e., adding T4 and adjusted daily difference between T2 and T4 to get T2). If PVD of T3 is greater than 5%, we estimated T2 by T3 using the shift method (i.e., adding T3 and adjusted daily difference between T2 and T3 to get T2).*" (line 111-116).

**Line 118:** A brief introduction to the "shift methods" is needed here.

**Response:** Thank you for your suggestion. We have added description of the shift method in the

revised manuscript.

"*Specifically, we used the shift method because there is non-linear relationship between daytime and nighttime LSTs (i.e., T2 and T3 (or T4)) (Crosson et al., 2012). We estimated the daily shift using temporally interpolating monthly averaged shift, i.e., monthly mean LST difference between T2 and T3 (or T4), and then we added the daily shift to T3 (or T4) to estimate T2.*" (line 119-122).

**Line 123:** Does "the overall mean of observations in each pixel" mean the pixel with all the four valid observations?

**Response:** Thank you for your question. "The overall mean of observations in each pixel" was the fitted daily values (temporal trend) in a year using the smoothing spline function for which the independent variable is the day of year. For example, the overall means of T2 and T4 were independently estimated. We have clarified it in the revised manuscript.

"*First, we fitted the overall mean of observations in each pixel (i.e., the fitted daily values (temporal trend) in a year using the smoothing spline function for which the independent variable is the day of year) using a smoothing spline function (Green and Silverman, 1994) to capture the overall trend. Specifically, the overall means of T2 and T4 were independently estimated.*" (line 126-129).

**Lines 127-128:** I have two confusions here. First, it remains unclear which type (e.g., linear regression or others) of the correlation has been used and how many of the neighboring valid pixels (or how large the neighborhood size) have been used for each target pixel. Second, if the value for the target pixel is missing, how did this study obtain the correlation between the target pixel and its neighboring valid pixels?

**Response:** Thank you for your questions. (1) We used linear regression correlations between target pixel and its 8 neighboring valid pixels. (2) We used the daily residuals of a year from target pixel and its neighboring pixels to estimate the missing values. So, when the value of the target pixels is missing for a specific day, we can still build linear regression functions based on the time series data. We have clarified it in the revised manuscript.

"*Second, we spatiotemporally interpolated residuals for each day using a correlation-based method (Details in Sect. 3.2.1), in which the missing residual of a target pixel was estimated based on the temporally and linearly regressive correlation between target pixel and its 8*

*neighboring valid pixels (i.e., with good quality). We used the daily residuals of a year from target pixel and its neighboring pixels to estimate the missing values. When the value of the target pixels is missing for a specific day, we can still build linear regression functions based on the time series data.*" (line 132-136).

**Line 137:** "In each of these years, we selected 19 days with the largest coverage of high quality data (coverage > 95% quantile)": It remains unclear what does the coverage mean? the coverage of all the four observations or the "overall mean" in a day ?

**Response:** Thank you for your question. The "coverage" means the percentage of the valid observations of an image at a specific time point of a day. We have clarified it in the revised manuscript.

"*In each year, we selected 19 days with the maximum observations of high quality data (i.e., daily data with valid observations larger than 95% percentile in a year) in the cross validation.*" (line 189-190).

**Lines 160:** Besides the contrasting performances between urban and rural areas, the accuracies may vary substantially by the climate zones due to the different data availability and seasonality. For example, the tropical regions may have lower accuracies than the high latitude regions. I am wondering if possible to show the model performances for difference climate zones.

**Response:** Thank you for your suggestion. We have added a table on the accuracies of gap-filled LST in different climate zones (Table S1) and discussed it in the revised manuscript. We found that accuracies vary in different climate zones and may be also correlated with landforms.

"*Besides, the accuracies of gap-filled LST vary with climate zones and may be also correlated with landforms (Table S1).*" (line 221-222).

**Table S1. Average root mean square error (RMSE) (Unit: °C) in excluded area in climate zones**

| Time | Climate zone | Landforms | Original missing rate (%) | RMSE (±standard deviation) | | |
|---|---|---|---|---|---|---|
| | | | | 25% | 50% | 75% |
| Daytime | Equatorial climate | Plain, Plateau, Mountain | 24.1 | 1.97 (±0.47) | 1.96 (±0.42) | 2.03 (±0.43) |
| | Arid climate | Desert, Plateau, Basin | 1.3 | 1.40 (±0.51) | 1.42 (±0.48) | 1.64 (±0.53) |
| | Warm temperate climate | Plain, Plateau | 13.7 | 1.81 (±0.50) | 1.79 (±0.46) | 2.05 (±0.59) |
| | Snow climate | Plain, Mountain | 16.2 | 2.17 (±0.66) | 2.25 (±0.68) | 2.50 (±0.74) |
| | Polar climate | Plateau of Tibet | 10.7 | 2.71 (±0.77) | 2.73 (±0.77) | 2.87 (±0.72) |

| Nighttime | Equatorial climate | Plain, Plateau, Mountain | 23.0 | 1.24 (±0.35) | 1.21 (±0.30) | 1.33 (±0.33) |
| | Arid climate | Desert, Plateau, Basin | 2.0 | 1.03 (±0.30) | 1.19 (±0.35) | 1.30 (±0.41) |
| | Warm temperate climate | Plain, Plateau | 10.5 | 1.11 (±0.27) | 1.23 (±0.32) | 1.40 (±0.42) |
| | Snow climate | Plain, Mountain | 13.1 | 1.80 (±0.57) | 1.81 (±0.64) | 1.97 (±0.64) |
| | Polar climate | Plateau of Tibet | 2.2 | 1.50 (±0.28) | 1.60 (±0.29) | 1.62 (±0.24) |

Note: Each root mean square error (RMSE) is the mean of RMSEs from all selected days in selected MODIS tiles of a specific climate zone (from 15 MODIS tiles).

**Line 179:** I think we cannot see the UHI phenomenon from figure 6 if we do not know where the urban areas are.

**Response:** Thank you for your comment. We have added urban boundaries in figure 6 to show UHI phenomenon.

**Line 179:** It would be much better if this study can summarize the existing seamless LST data by a table.

**Response:** Thank you for your suggestion. We have added a table for the existing seamless LST datasets covering a large spatial extent in the supplement (Table S2) and discussed them in the introduction and discussion in the revised manuscript.

**Table S2. Summary of existing seamless LST dataset at large spatial extent**

| Literature | Spatial extent | Spatial resolution | Temporal frequency | Coverage time | Accuracy (default: RMSE) |
|---|---|---|---|---|---|
| Metz et al. (2017) | Global | 3 arc-min (~5.6 km at the equator) | Monthly | 2003 – 2016 | 0.5K |
| Li et al. (2018) | United States, urban and surrounding areas | 1 km | Daily (mid-daytime and mid-nighttime) | 2010 | 3.29°C (day), 2.68°C (night) |
| Zhao et al. (2020) | China | 1 km | Monthly | 2003 – 2017 | 1.59°C |
| Cheng et al. (2021) | China | 1 km | Daily (mid-daytime and mid-nighttime) | 2002 – 2020 | 3K |
| Zhang et al. (2021b) | China and surrounding areas | 1 km | Daily, all weather | 2000 – 2020 | 2.03K to 3.98K |
| Zhan et al. (2021) | Global | 1 km | Daily, Average | 2003 – 2019 | Mean absolute error is 1.0K |
| Shiff et al. (2021) | Global (need to run code on GEE) | 1 km | Daily (mean, mid-daytime and mid-nighttime) | 2002 – 2020 | 2.7°C (mean) |

| This study (Zhang et al., 2021a) | Global | 1 km | Daily (mid-daytime and mid-nighttime) | 2003 – 2020 | 1.83°C (day), 1.28°C (night) |
| --- | --- | --- | --- | --- | --- |

**Reference**

[revised manuscript text omitted]

**CC #1:**

Dear Dr. Zhou,

It's a great pleasure for me to participate in this open discussion. This paper proposed clearly significant work for improving the practicality of the current MODIS LST product. As my phD dissertation is also related to cloudy-sky LST estimation and LST data applications, this work highly attracts me and I believe if the released dataset achieves satisfactory accuracy and data quality, it will definitely help the communities in monitoring surface thermal dynamics, .etc.

I do have several questions and concerns that I cannot fully understand from the paper, therefore, I would like to list them below, and would you please have a clarification?

Thank you for your work and contributions to the topic. Also, I sincerely appreciate your time for explaining my questions.

Warm regards,

Aolin Jia
PhD candidate
Department of Geographical Sciences
University of Maryland, College Park, MD, USA

**Major Questions:**

**1)** the top question I would like to ask is related to the major comment (4) from Referee #1. The assessment method of the proposed work only focused on clear-sky samples, whereas recovered cloudy-sky results are the main contributions of this study that should be focused on.

This study assessed the results by hypothetically removing some clear-sky days and filling them, and then the filled results were compared with the official MODIS LST. However, not like the shortwave variables (surface reflectance), LST values under clouds are affected by the cloud coverage (cooling/cooling effect at daytime/nighttime), thus clear-sky samples are not representative for the cloudy-sky cases.

Besides, the hypothetically removed clear-sky pixels usually have enough clear-sky adjacent pixels for interpolation, and they can easily get good interpolation accuracy than actual cloudy cases, which might be not fair to only use this assessment method to demonstrate the accuracy in cloudy-sky. Moreover, this assessment method cannot involve the filled LST samples where official MODIS has bad data quality (partially cloud-covered or large view zenith angle), while they are also important parts of the contributions of this work.

Therefore, I am really curious about the accuracy validated by ground measurements that can comprehensively assess both clear-sky and cloudy-sky samples, independently. This work has mentioned that site measured record is not comparable to remote sensing retrieved LST due to spatial scale mismatch or broadband emissivity (BBE) uncertainty. In fact, those issues have been discussed in previous studies. Li et al. (2021) have proved that SURFRAD sites have little heterogeneity issue for validating 1-km LST product by analyzing Landsat LST, and BBE would not introduce noticeable errors for the validation (Xing et al., 2021). And that's why most related studies in this topic used ground measurement for assessment (Zhang et al., 2021, Xu and Cheng, 2021, Zeng et al., 2018).

The proposed work followed Li et al. (2018) and didn't consider the site validation, but I think this is mainly because Li et al. (2018) only focused on US urban areas that there is no available ground LST or upward longwave radiation measurements, besides, the heterogeneity issue in urban areas is substantial and cannot be ignored. As this study generated the global gap-free LST, it would be a great opportunity to assess the product by using high-quality site measurements, such as SURFRAD and BSRN, globally.

It is reasonable that the site-validated RMSE is not comparable to the cross-validation results, but by separately validating clear-sky and cloudy-sky results using ground site measurement and comparing the RMSEs in clear days and cloudy days, this study can demonstrate the accuracy stability and real interpolation accuracy of the proposed product, which I believe will be what users care about.

**Response:** Thank you very much for your suggestion. We mainly obtained clear-sky LST data using the spatiotemporal fitting algorithm in this study. Therefore, the cross-validation method is enough to evaluate the performance of the algorithm, and the performance of the original LST data was already evaluated using ground site measurements. We understand that our clear-sky LST data are different from the all-weather LST that can be further estimated based on the seamless clear-sky LST data from this study by adding the LST differences resulting from cloud

impacts according to the surface energy balance (SEB) methods (e.g., Jia et al., 2021). We clarified it in the revised manuscript.

"*A limitation of this study is that the gap-filled LST dataset mainly reflects the clear-sky conditions and future work can focus on recovering cloudy-sky LST to produce all-weather LST dataset when high-quality ancillary data become available. As only the spatiotemporal information of clear-sky MODIS LST data was used to fill the missing values, the gap-filled pixels mainly reflect the clear-sky LST and might overestimate the actual LST values. Previous studies have attempted to develop methods for obtaining all-weather LST data by incorporating cloudy-sky LST retrieved from passive microwave observations or reanalyzed products (Duan et al., 2017; Long et al., 2020; Zhang et al., 2019, 2020, 2021b) or adding clear-sky LST and the LST differences resulting from cloud impacts according to the surface energy balance (SEB) methods (e.g., Jia et al., 2021). However, it is challenging to obtain cloudy-sky LST and cloud caused LST differences at the global scale in the last two decades because the ancillary datasets have lower spatial resolutions and accuracies compared to MODIS LST, leading to complicated algorithms with complicated hypotheses (Long et al., 2020; Zhang et al., 2021b). Future studies are needed to develop robust and efficient algorithms for producing global all-weather LST data.*" (line 318-328).

**2)** I'm still a little bit confused about the methodology. Would you please use plain language to explain how the interpolation methodology captures the day-to-day variation signals where there are cloud covers with considerable spatiotemporal scales, such as raining seasons (continuity could be weeks) in southern China or tropical areas without introducing passive microwave or modeling data?

**Response:** Thanks for your question. When there are continuously days with clouds in southern China, the high-quality MODIS observations (clear-sky LST when the land surface is still wet) can capture low values. The smoothing spline function can capture this temporal trend since this algorithm does not have any hypothesis on the shape of the seasonal trend. Pixels in southern China have similar temporal pattern of LST during this raining period. The residuals between MODIS LST and fitted trend are linearly correlated with each other in local regions based on the daily time series residual data. Therefore, we can build functions of residuals between target pixel and its neighboring pixels. When the residual of a specific day at the target pixel is missing, it can be estimated using the residual values of the neighboring pixels and the built functions.

Therefore, the missing LST values can be estimated by adding the fitted temporal trends and the estimated residuals. We have improved the description in Section 3.2 of the revised manuscript.

**3)** Interpolation-based method usually has a trade-off between calculation efficiency and accuracy because essentially they used statistical relationships with referred pixels, the more referred pixels they can obtain, the higher accuracy they will achieve, but more time will be consumed. This category of the methodology has been well developed and would you please strengthen the breakthrough of this proposed work, and what are its differences from previous interpolation methods, and how could the efficiency be improved while high accuracy is maintained?

**Response:** Thanks for your question. The method we used is not a new interpolation technique, and we mainly improved the computing efficiency for large scale application purpose. We have clarified it in the revised manuscript. Moreover, we focused on the development of a new dataset instead of a new interpolation method in this paper.

"*Moreover, 1% of pixels at central pixels of blocks (10 × 10 pixels) were used as neighboring pixels for interpolation of residual (Sect. 3.2.2) to reduce the amount of calculation, and the relevant parameters (i.e., correlation coefficients and coefficients of linear regression function) between target pixel and its 8 nearest neighboring pixels were calculated only one time for the entire period (365 days for a year) based on the time series of residuals. The reason is that the time series of residuals from two neighboring pixels within a short distance are highly correlated with each other.*" (line 306-310).

**4)** I also have a question about neighboring pixels in a spaital window: it looks like land cover type and elevation differences were not considered in the method (maybe I missed). Land cover types highly impact the LST values even they have close spatial distance. For example, forest LST is the thermal signal from the canopy, but neighboring LST of open grassland could be from the ground (even soil surface), which are significantly different. That's why Zhang et al. (2021) build models by land cover types in each window. Jin and Treadon (2003) also build typical diurnal temperature cycles for each land cover type. May I ask if you have included such consideration? If not, could you please have some discussion to clarify that those differences won't significantly affect the interpolation accuracy?

**Response:** Thank you again for your questions. We did not use land cover type data or elevation

data in this study, but the accuracy of the gap-filled LST in this study would not be significantly affected by land cover types and elevations. We have added a paragraph to discuss it in the revised manuscript.

*"The accuracy of the gap-filled LST should not be significantly affected by the land cover type and elevation differences in local spatial windows. LST values from different land cover types and elevations within a small spatial region may be significantly different (Zhang et al., 2021b). These differences of LST values can be captured though the temporal pattern of LST (overall mean) by separately fitting the smoothing spline curves (Fig. 3), and the spatiotemporal similarity of residuals between neighboring pixels were gap-filled. The gap-filled LST values are the sum of overall mean and residuals. Therefore, our method can capture the missing values of LST in different land cover types and elevations in local spatial windows."* (line 312-317).

**Other Comments:**

I also have some minor comments that might be helpful for the paper publication.

1) I found some typos and grammar errors in the manuscript so would you please revise them? These are not all of them so I would suggest that English editing would improve the writing quality.

Line 11: parameter -> parameters

**Response:** done!

Line 35: measured is not accurate, could be retrieved

**Response:** done!

Line 40: ASTER -> Advanced Spaceborne Thermal Emission and Reflection Radiometer (ASTER)

**Response:** done!

Line 47: "a" is missing before "larger number"

**Response:** done!

Line 50: "a" is missing before "daily"

**Response:** done!

Line 57: "the" is missing before conterminous US.

**Response:** done!

Line 65: relationships

**Response:** done!

Line 71: "a" is missing before hybrid method

**Response:** done!

Line 83: "the" is missing before "methods mentioned above"

**Response:** done!

Figure 1: would you please add H and V numbers to the x-axis and y-axis of Figure 1

**Response:** done!

Line 102: first ")" could be removed

**Response:** It should not be removed. It is for Sect. 3.1 in "data pre-processing (Sect. 3.1)".

Line 115: "If PVD of T2 >= 5%, did not change the data and accepted it" would you please explain this sentence? and why 5%?

**Response:** Thank you for your question. We have clarified in the revised manuscript.

"*We selected the threshold of PVD as 5% because the valid data smaller than 5% is not enough to capture the spatial pattern of LST in a tile according to our experiments.*" (line 117-119).

Line 138: 'the' is missing before selected days

**Response:** done!

Line 139: 'originally values' -> 'original values'

**Response:** done!

Line 158: numbers, 'are' -> being

**Response:** done!

Line 175: in day 200 -> on

**Response:** done!

Line 182: are -> is

**Response:** done!

Line 210: exist

**Response:** done!

Line 218: has -> have

**Response:** done!

Line 254: 'those' could be removed?

**Response:** done!

2) in the supplementary materials, would you please explain why this method refers to center pixels from neighboring blocks for interpolation (could be 10 km far from the target pixel) rather

than using neighboring pixels. S1 is easily understood but could you have a clarification on why the "S2 Implementation of the ICW method" is designed like this? Thanks!

**Response:** Thank you for your question. We have moved S1 and S2 to Section 3.2 and changed them as Section 3.2.1 and Section 3.2.2. Details are shown in line 137-180 in the revised manuscript. We have clarified it in the revised manuscript.

"*We selected 1% of the uniformly distributed pixels (10 km intervals) as representative neighboring pixels to perform the interpolation of residuals with high efficiency without reducing the accuracy based on our experiments. Moreover, we divided the global land surface area into 9 overlapped zones to avoid possible boundary effects (Details in Sect. 3.2.2). Finally, the seamless overall mean and daily residuals were added to obtain the gap-filled LST data.*" (line 136-140).

"*The reason is that the time series of residuals from two neighboring pixels within a short distance are highly correlated with each other. Our scheme can significantly improve the efficiency for global applications without reducing the accuracy according to our experiments.*" (line 309-311).